# Impact of Copper-Doped Mesoporous Bioactive Glass Nanospheres on the Polymerisation Kinetics and Shrinkage Stress of Dental Resin Composites

**DOI:** 10.3390/ijms23158195

**Published:** 2022-07-25

**Authors:** Danijela Marovic, Matej Par, Tobias T. Tauböck, Håvard J. Haugen, Visnja Negovetic Mandic, Damian Wüthrich, Phoebe Burrer, Kai Zheng, Thomas Attin, Zrinka Tarle, Aldo R. Boccaccini

**Affiliations:** 1Department of Endodontics and Restorative Dentistry, University of Zagreb, 10000 Zagreb, Croatia; vnegovetic@sfzg.hr (V.N.M.); tarle@sfzg.hr (Z.T.); 2Department of Conservative and Preventive Dentistry, Centre for Dental Medicine, University of Zurich, 8032 Zurich, Switzerland; tobias.tauboeck@zzm.uzh.ch (T.T.T.); damian.wuethrich@uzh.ch (D.W.); phoebe.burrer@zzm.uzh.ch (P.B.); thomas.attin@zzm.uzh.ch (T.A.); 3Department of Biomaterials, Institute of Clinical Dentistry, University of Oslo, 0317 Oslo, Norway; h.j.haugen@odont.uio.no; 4Engineering Research Center of Stomatological Translational Medicine, Nanjing Medical University, Nanjing 210029, China; kaizheng@njmu.edu.cn; 5Department of Materials Science and Engineering, Institute of Biomaterials, University of Erlangen-Nuremberg, 91058 Erlangen, Germany; aldo.boccaccini@fau.de

**Keywords:** mesoporous, copper, polymerisation, light transmission, polymerisation kinetics, linear shrinkage, polymerisation shrinkage stress, depth of cure

## Abstract

We embedded copper-doped mesoporous bioactive glass nanospheres (Cu-MBGN) with antibacterial and ion-releasing properties into experimental dental composites and investigated the effect of Cu-MBGN on the polymerisation properties. We prepared seven composites with a BisGMA/TEGDMA (60/40) matrix and 65 wt.% total filler content, added Cu-MBGN or a combination of Cu-MBGN and silanised silica to the silanised barium glass base, and examined nine parameters: light transmittance, degree of conversion (DC), maximum polymerisation rate (R_max_), time to reach R_max_, linear shrinkage, shrinkage stress (PSS), maximum PSS rate, time to reach maximum PSS rate, and depth of cure. Cu-MBGN without silica accelerated polymerisation, reduced light transmission, and had the highest DC (58.8 ± 0.9%) and R_max_ (9.8 ± 0.2%/s), but lower shrinkage (3 ± 0.05%) and similar PSS (0.89 ± 0.07 MPa) versus the inert reference (0.83 ± 0.13 MPa). Combined Cu-MBGN and silica slowed the R_max_ and achieved a similar DC but resulted in higher shrinkage. However, using a combined 5 wt.% Cu-MBGN and silica, the PSS resembled that of the inert reference. The synergistic action of 5 wt.% Cu-MBGN and silanised silica in combination with silanised barium glass resulted in a material with the highest likelihood for dental applications in future.

## 1. Introduction

Research on ion-releasing dental restorative materials is expanding. The term ‘bioactivity’ has increasingly been used in connection with dental restorative materials in original research over the past decade [1,2], and various attempts have been made to produce a strong and durable material with bioactive properties [3,4,5,6,7].

The failure of direct dental restorations caused by secondary caries and fractures [8] is the main incentive for this increased research. Almost 60% of all restorations are replacements for previous restorations due to secondary caries [9,10], which significantly increase costs for dental care [11]. Conventional composite materials lack the necessary acid-buffering capacity and antibacterial properties [11,12,13] to prevent secondary caries, especially in individuals at high caries risk.

Secondary caries due to resin composite materials is, at least in part, a material-related problem [12]. An inherent flaw of composites is polymerisation shrinkage (PS), which is inevitably related to polymerisation processes. PS results from a significant reduction in the intermolecular distance between monomers due to the formation of covalent bonds during free-radical polymerisation. Under unconstrained conditions, PS has no adverse effects. Nevertheless, composites are bonded to the tooth structure, often in a confined cavity. During polymerisation, PS is accompanied by a gradual development of elastic modulus, which increases the stiffness of the material. However, the adhesive bond simultaneously limits the contractile adaptation of the composite. This places stress on the cavity walls, which is referred to as polymerisation shrinkage stress (PSS) [14].

A further increase in PSS can lead to cusp flexion and enamel or restoration microcracking and debonding at the restoration–tooth interface [15]. This discontinuity at the restoration–tooth interface results in marginal gaps or internal voids. Marginal gaps larger than 250–400 µm contain significant numbers of bacteria that are protected from mechanical removal by tooth brushing and exposed to oral fluids rich in the nutrients they need to survive and multiply [16,17]. Such marginal gaps often predict secondary caries [16,17,18]. Thus, PSS is associated with secondary caries, although a definite clinical relationship has not yet been established [15].

Experimental resin composites containing 15 wt.% bioactive glass (BG) reduced bacterial penetration in narrow 15–20 µm wide marginal gaps [19]. This effect was explained by the alkalising properties of BG and the creation of unfavourable conditions for bacterial growth in the gaps. Moreover, the apatite crystals formed in the marginal gaps restricted the physical penetration of bacteria into deeper areas. The intrinsic antibacterial properties of BG can be further enhanced by adding antimicrobial substances to the porosity of mesoporous BG particles [1,20].

Although the antimicrobial activity of the well-known 45S5 BG and of some other BGs is remarkable, caution is required when making generalised statements, given the variety of different BG compositions, particle sizes, surface modifications, and volume fractions in the final products [21]. Zheng et al. developed copper-doped mesoporous BG nanospheres (Cu-MBGN) using a microemulsion-assisted sol–gel method [22]. Spherical particles with a diameter of 100–300 nm and porosity of 2–10 nm were prepared and doped with 0–6 mol.% Cu^2+^. The particles released copper, calcium, and silica, and proved the formation of apatite in vitro [22]. Copper is known for its bactericidal effects [23], since it accumulates in and disrupts the bacterial membrane, has an oxidising effect, and inhibits replication [24]. Therefore, the introduction of copper-doped particles could be an interesting approach to suppressing bacterial activity in marginal gaps and counteracting secondary caries.

Commercially available dental resin composites are heterogeneously filled systems containing various glass, silica, and quartz fillers [25]. Large and small fillers are combined to achieve the desired properties; for example, a high filler content improves mechanical properties [25], whereas a low filler content with larger particle sizes (>20 μm) increases light transmittance [26], which is desirable for bulk-fill composites. For new fillers, polymer systems are usually tested with single fillers [27,28]. Although this approach allows a thorough characterisation of the contribution of individual components on the overall composite behaviour [27], systems with single fillers are unlikely to be used for the final products that reach the market. Moreover, the interaction between fillers should be considered.

We recently developed experimental dental resin composites with Cu-MBGN to investigate the possibility of introducing Cu-MBGN into hybrid resin composites as commercial materials. Cu-MBGN with a large surface area have high reactivity and ion release capability [22]; hence, adding Cu-MBGN in small amounts (1–10 wt.%) is possible. Furthermore, inert silica fillers have been found to promote ion release [3] and serve as nucleation sites for crystallisation [3,27,29]. Adding Cu-MBGN improved the flexural strength and microhardness of the composites. Water immersion for 28 days did not reduce the flexural strength of experimental materials containing combined silanised silica particles and Cu-MBGN, but such stability was not observed for materials containing only Cu-MBGN [5].

This study continued our previous work [5], which aimed to investigate the effect of Cu-MBGN on composite properties related to polymerisation. To our knowledge, no comparable study has investigated the influence of mesoporous particles on the shrinkage and shrinkage stress of dental composites. As in our previous study [5], we doped resin composites with Cu-MBGN in complex binary or ternary filler systems and standardised BisGMA/TEGDMA resin and inert silanised Ba-glass micro fillers (1 μm) for all composites.

For the binary systems, we wanted to investigate the effects of adding 10 wt.% Cu-MBGN and compare them with the effects of similar amounts of reference fillers: 10 wt.% silica (12 nm) as the inert and 10 wt.% BG 45S5 (4 μm) as the bioactive reference material. For ternary systems, we studied the combined effect of silica nanofillers and Cu-MBGN using amounts of 1 wt.% or 5 wt.% Cu-MBGN and adding silica up to 14 wt.%. The reference materials resembled the binary systems and contained only two filler types: 14% silica or 14% BG 45S5. The null hypotheses were as follows:(1)Adding Cu-MBGN has no effect on light transmission, degree of conversion, maximum polymerisation rate, time to reach maximum polymerisation rate, linear shrinkage, shrinkage stress, maximum shrinkage stress rate, time to reach maximum shrinkage stress rate, and depth of cure.(2)There is no difference between composites containing only Cu-MBGN and those containing combined Cu-MBGN and silica of 12 nm particle size in any of the examined parameters.(3)There is no difference between Cu-MBGN and the conventional 45S5 BG of 4 μm particle size in any of the examined parameters.

## 2. Results

### 2.1. Light Transmittance

The 10-Si sample had the highest light transmittance, whereas 5-CuBG-Si had the lowest transmittance, as shown in Figure 1. Adding Cu-MBGN reduced light transmittance considering that all Cu-MBGN-containing materials had significantly lower light transmittance values than the materials without Cu-MBGN. Furthermore, increasing the amount of either silica or 45S5 BG further diminished their respective light transmittances. Figure 2 depicts the development of light transmittance during 20 s of light irradiation. Unlike the other tested materials, 10-CuBG demonstrated an increase in light transmittance during the first second, followed by a drop in the third second, and a relatively stable light transmittance to the end of irradiation. Other materials showed an increase in light transmittance during the same period.

### 2.2. Polymerisation Kinetics

Figure 3 shows that the highest DC was achieved by 10-CuBG sample (58.8 ± 0.9%) in the binary group, which was statistically higher than in the inert control 10-Si group (54.7 ± 0.5%) or the bioactive reference 10-BG with 45S5 BG (51.5 ± 0.3%). We observed similar behaviour for the maximum reaction rate.

In the ternary group, adding Cu-MBGN and silica fillers produced a dose-dependent decrease in DC. Although adding 1% Cu-MBGN to the 1-CuBG-Si did not influence the DC or maximum reaction rate, the 5-CuBG-Si composite showed a 4% reduction in DC (53.1 ± 0.5%) and a 5% reduction in the maximum reaction rate (7.8 ± 0.2%/s) compared to the reference material 14-Si (57.5 ± 0.3% and 8.2 ± 0.2%/s).

However, the BG composites (10-BG and 14-BG) had the lowest DC and the lowest maximum reaction rate in both the binary and ternary groups. 14-BG had the lowest DC (51.1 ± 0.7%) and the lowest maximum reaction rate (5 ± 0.1%/s).

There was no difference in the time required to achieve the maximum reaction rate among any of the tested materials (3.1–3.4 s).

Table 1 shows the fit parameters for the exponential sum function y = a × (1 − e^−bx^) + c × (1 − e^−dx^). Parameters ‘a’ and ‘b’ denote the gel phase of the polymerisation, while parameters ‘c’ and ‘d’ explain the glass phase [29]. Parameter ‘a’ followed the general pattern of the DC and R_max_ values. Adding Cu-MBGN fillers in the binary group increased all the parameters compared to the inert control 10-Si and bioactive control 10-BG. The combination of Cu-MBGN and silica fillers in the ternary group diminished the ‘a’ parameter. However, parameters ‘b’ and ‘d’ showed an inverse relationship in the ternary group, effectively increasing the values for 1-CuBG-Si and 5-CuBG-Si. Parameter ‘c’ increased only for 1-CuBG-Si compared to 14-Si.

### 2.3. Linear Shrinkage

Figure 4 shows that the highest linear shrinkage was for the inert control material 10-Si (3.12 ± 0.08%) and 1-CuBG-Si (3.07 ± 0.06%), whereas the control bioactive material 14-BG had the lowest values (2.45 ± 0.05%). However, 10-CuBG had the quickest shrinkage, followed by 1-CuBG-Si. At the 20 s level, 10-Si reached its shrinkage level and remained the material with the greatest shrinkage.

### 2.4. Polymerisation Shrinkage Stress

Figure 5 shows the comparisons of the end values for PSS. A group of materials 1-CuBG-Si (0.97 ± 0.06 MPa), 10-CuBG (0.89 ± 0.07 MPa), and 10-Si (0.84 ± 0.13 MPa) had the highest PSS. The bioactive reference materials (14-BG and 10-BG) had the lowest PSS (0.52 ± 0.06 MPa and 0.69 ± 0.09 MPa, respectively). These bioactive reference materials also had the lowest maximum PSS rate and the slowest shrinkage stress development.

The 10-CuBG composite, followed by both CuBG composites from the ternary group of composites (1-CuBG-Si and 5-CuBG-S), demonstrated the maximum PSS rate. This behaviour was reinforced by the fact that the maximum PSS rate for these materials was 2–3 times shorter than for the bioactive reference materials 10-BG and 14-BG. 10-CuBG had the quickest PSS rate (in the first 10 s after the start of irradiation), whereas, after the 10th second, 1-CuBG-Si became the material with the highest PSS (Figure 6).

### 2.5. ISO 4049 Depth of Cure

The greatest depth of cure was achieved by the reference material 10-Si, at 3.6 ± 0.1 mm (Figure 7). A medium-level depth of cure (3.2–3.3 mm) was attained for a group of materials: 10-BG, 1-CuBG, 14-Si, and 14-BG. The shallowest depth of cure was exhibited by 10-CuBG and 5-CuBG-Si, with values of 2.6 ± 0.03 and 2.7 ± 0.05 mm, respectively.

## 3. Discussion

To our knowledge, this is the first in-depth analysis of the polymerisation, shrinkage, and stress kinetics associated with the incorporation of nanoscale mesoporous particles into a polymer network. We found different behaviours in the binary systems consisting of inert micro fillers and Cu-MBGN fillers and in the ternary blend combining Cu-MBGN and silica fillers. Adding Cu-MBGN to the binary filler blend enhanced DC at a clinically relevant 2 mm depth, although the violet–blue light transmission decreased. However, the high light transmission through the binary 10-CuBG in the first 3 s after the onset of illumination resulted in the highest reaction rate. Consequently, adding Cu-MBGN to 10-CuBG increased the PS and PSS rate in the initial phase of the polymerisation reaction, although the final values remained the same as for the inert reference. In contrast, the combination of Cu-MBGN and silica in ternary composites showed a dose-dependent decrease in the polymerisation rate and DC. This behaviour led to an increase in PS in the 1-CuBG-Si sample. However, the PSS of binary 10-CuBG and ternary 5-CuBG-Si remained at the same level as the inert controls. Thus, all three null hypotheses were rejected.

### 3.1. Binary Composites

Adding Cu-MBGN apparently caused less attenuation than the other two reference materials in the binary group. The effective light transmittance through a 2 mm depth was 24% of incident light for 10-CuBG, but 49% and 39% for the inert and bioactive controls, respectively. In a simplified model, light transmission was determined by the amount of reflected, absorbed, and scattered light. Although some reflection from the specimen surface could occur [30], precautions were taken to minimise light reflection by keeping the light guide tip perpendicular to the specimen’s surface [31]. Furthermore, most of the reflection occurred at the interface between air and the PET foil on top of the specimen, which was identical for all materials. Hence, the effect of light reflection was consistent throughout the light transmittance measurements and introduced minimal experimental variability.

Scattering in dental resin composites depends on filler loading, particle size, morphology, and proximity of the refractive indices of resin and fillers [32]. At the same filler loading, nanosized particles have a larger surface area than microparticles and, thus, a larger area where light reflection can occur [33]. Mesoporous particles are characterised by even larger and more porous surface areas, which causes higher scattering frequencies or diffuse light reflection [27,34]. For this reason, they are often used for the fabrication of anti-reflection coatings [35] or solar cells [36]. Cu-MBGN is reported to have a particle size of about 100 nm [5] and a large specific surface area of 317 m^2^/g with pore sizes of 2–10 nm [22]. This is almost twice the surface area of the silica fillers used in this study (160 m^2^/g). Theoretically, the rough surface of Cu-MBGNs could direct photons in multiple directions, resulting in diffuse backscattering and enhanced light absorption [37] by photoinitiators in the surrounding resin matrix.

Although average effective transmittance values (over 20 s of illumination) are given here, it is important to understand polymerisation as a dynamic process in which some parameters constantly change during and after light exposure. The rate of polymerisation is fastest in the quasi-static phase [38], in the first few seconds (3–4 s in our study) after the onset of irradiation, when most of the monomer radicals react with other monomers and the chain length increases [29]. After the gel point, the reaction rate decreases by several orders of magnitude due to limitations in diffusion coefficients and continues gradually after curing [39]. The viscosity and elastic modulus increase during polymerisation, and the resin matrix changes its refractive index as the monomers convert into polymer chains [40]. Fillers, however, do not change their refractive indices. When the refractive index of the matrix approaches that of the fillers, the translucency of the composite increases, but the transmission efficiency decreases [29]. The mismatch of the refractive indices of resin and filler determines the opacity of the material [40].

The light transmission of most commercial composites increases during and after polymerisation, which allows light penetration into deeper layers and the activation of photoinitiators [29]. The same behaviour was found for all the materials studied, except 10-CuBG. In the case of 10-CuBG, the light transmission was higher in the first 3 s and increased up to 1 s, after which it gradually decreased by 15% and reached the second lowest light transmittance value. This behaviour was unique. For other materials, the light transmittance increased by 81% (10-BG) and 97% (10-Si) over the 20 s curing time. The refractive indices of the unpolymerised BisGMA/TEGDMA monomer blend and Ba-glass micro fillers were almost identical, amounting to 1.52–1.53 [41]. The refractive index of the polymerised BisGMA/TEGDMA comonomers increase to 1.55–1.56 [41]. In this study, we did not measure the refractive index of Cu-MBGN due to technical limitations. However, on the basis of the Appen factors and the equation provided by Tiskaya et al. [2] for dense particles, we calculated a value of 1.44. A study on mesoporous niobium silicate particles showed that the refractive index of the particles in the BisGMA/TEGDMA base was 1.43–1.45 [27]. Therefore, it is likely that the decrease in light transmittance of 10-CuBG was determined by the increase in opacity due to the increase in the refractive index upon polymerisation of the resin base.

This theoretical consideration was confirmed by the fastest polymerisation reaction rate of 10-CuBG, consistent with the high light transmission in the first 3 s and the time of 3.5 s to reach the maximum reaction rate. Most of the polymerisation apparently occurred in the gel phase, although all polymerisation kinetics parameters for 10-CuBG were highest in the binary group. However, the increased transmission in the first second could not be the only reason for the highest DC. All reference materials (inert and bioactive) exhibited higher transmission at the same time.

One possible reason for this could be that diffuse light reflection from the mesoporous Cu-MBGN structure activated more photoinitiator molecules at the beginning of irradiation. A similar behaviour was found in the only other study known to the authors that investigated the polymerisation kinetics of composites containing mesoporous niobium–silica particles [27]. Despite the much lower refractive index of 1.43–1.45, the niobium silicate composite achieved a higher DC and a faster polymerisation rate than the barium borosilicate glass-based control composite [27]. The ‘a’ and ‘b’ parameters of 10-CuBG were higher than those of 10-Si, indicating a faster onset of polymerisation and a faster transition to the gel phase of 10-CuBG. The magnified plot of the evolution of PS in the first 20 s (Figure 4C) and the variation of the polymerisation rate confirmed the mathematical calculations and showed rapid polymerisation and shrinkage. Regardless of the differences in the composition of niobium silicate and Cu-MBGN, their surface topologies are similar [27]; hence, we could assume similar behaviour. However, light scattering at the surface of Cu-MBGN particles combined with a short gel phase and low light transmittance must have contributed to the low depth of the cure, even at high DC.

High DC is usually directly correlated with high PS [42]. However, the PS of 10-CuBG was not the highest, despite 10-CuBG having the highest DC of all the materials tested. There are several explanations for this behaviour. First, the porosity of Cu-MBGN and the much larger particle size compared to silica could have physically prevented the resin from shrinking. The second reason could be the experimental setup, which recorded motion in the axial direction but allowed the material to shrink in the in-plane direction during the pre-gel phase [43].

Silanised fillers forming chemical bonds with the resin matrix may have caused internal stresses in the composite due to the different coefficients of thermal expansion and the ‘hoop stresses’ surrounding the fillers [44]. Condon and Ferracane showed that non-silanised micro fillers provide 30% stress relief compared to silanised fillers [45,46]. Meereis et al. hypothesised that spherical non-silanised fillers reduce PSS due to rotational and translational motions during polymerisation [47]. However, non-silanised fillers increase the viscosity of the material and increase the elastic modulus [46]. Our previous study on similar Cu-MBGN-containing composites confirmed the latter and showed the increased modulus of the 10-CuBG material [5], which had the same composition as the 10-CuBG used in this study. Although the Cu-MBGN particles were unsilanised and spherical, we did not observe the expected stress relaxation in this study, but the PSS values resembled those of the inert reference 10-Si. The porous surface of Cu-MBGN probably allowed the resin to penetrate and interlock after polymerisation, preventing rotational motion.

However, due to the lower PS of 10-CuBG, its PSS was aligned with that of the inert reference material 10-Si. The semi-rigid experimental setup in our study allowed a displacement similar to that of dental hard tissues [48] and provided a C-factor = 2. As for the PS measurements, the PSS setup measured the stresses generated in the post-gel phase [49], ensuring that they were comparable and relevant.

Real-time measurements of PS and PSS gave us insight into a specific behaviour of 10-CuBG that was also found for 1-CuBG-Si and 5-CuBG-Si in the ternary group. All these materials exhibited a faster initial evolution of PS and PSS than the other materials, which increased abruptly after the end of irradiation at 20 s (Figure 4C and Figure 6B), but to a much lesser extent in the case of 5-CuBG-Si. Indeed, the free-radical polymerisation of poly(dimethacrylate) is an exothermic reaction in which temperatures rise at high reaction rates [42], as observed for both 10-CuBG and 1-CuBG-Si. In conjunction with the heat generated by the light-curing device, we expected increased heat generation. The increased temperature in the samples decreased the resin density in the glass phase and stimulated the movement of monomeric radicals [42,50], which probably contributed to the increase in DC. At the same time, thermal expansion during light irradiation decreased PS and PSS, followed by an increase after irradiation.

### 3.2. Ternary Composites

Although the specific light transmittance determined the behaviour of the binary composites, the ternary composites were mainly affected by the increase in the particle surface area. As mentioned earlier, adding fillers without a coupling agent changes the rheology by increasing the viscosity of the composites [46]. The doubling of the surface area compared to the commonly used nanoparticles (Cu-MBGN vs. silica) further increased the viscosity. In the ternary group, all composites had paste-like consistency, like sculptable composites. In the binary group, 10-CuBG was also sculptable, whereas the reference materials had the consistency of flowable composites with the same filler weight.

Like 10-CuBG, reduced light transmission was also observed for 1-CuBG-Si and 5-CuBG-Si samples. Interestingly, there was no decrease in light transmittance with the progression of polymerisation, but there was an increase resembling that of the reference materials, although to a lesser extent. The combination of silica and Cu-MBGN, which is preferred because of the biomineralising effect and improved flexural strength and hardness [3,5], probably contributed to the increase in viscosity and light transmittance. Silica and Cu-MBGN particles are both silicate-based and have a considerable number of hydroxyl groups on their surfaces. They are both hydrophilic, whereas the resin is predominantly hydrophobic despite the hydroxide groups on BisGMA [42]. Due to the large specific surface area of Cu-MBGN particles, they are highly reactive. We hypothesise that intermolecular hydrogen or some other noncovalent bond formed between silica and Cu-MBGNs leads to the formation of supraparticles [20,51]. At least partial coverage of the surface of the Cu-MBGNs would result in lower diffuse light reflectance and altered light transmission behaviour. Moreover, when the size of these theoretical supraparticles reaches ~200 nm—half the wavelength of the activating light—the strongest scattering can be expected [52]. The expected concentration of supraparticles is larger in 5-CuBG-Si than in 1-CuBG-Si; thus, we can assume that light attenuation is more pronounced in the 5-CuBG-Si samples.

With analogous compositions, 14-Si and 1-CuBG-Si showed no differences in the final DC values or reaction rates. Nevertheless, the kinetic parameters ‘b’, ‘c’, and ‘d’ of 1-CuBG-Si were higher than those of 14-Si, indicating an earlier onset of gelation and immobilisation of the resin [53]. The parameters ‘c’ and ‘d’ describe the glass phase during radical polymerisation and are hardly influenced by external factors in commercial composites [38]. During the glass phase, all the conversion from double to single C–C bonds can be attributed to the elongation of the chain length [38]. As described earlier, the heat generated by the exothermic reaction and the curing unit slightly increased the translational and rotational radical end-mobility in the vitrified resin due to the thermally induced reduction in viscosity [42]. The thermal expansion decreased after light curing, increasing the PS and, consequently, the PSS. Unfortunately, thermal measurements were not part of this study, but they are planned for the near future.

As a result of decreased light transmission and increased viscosity, the maximum reaction rate of 5-CuBG-Si diminished. The DC value after 5 min was 4% lower than that of the inert reference material 14-Si DC. Initially, high viscosity hinders the propagation of radical polymerisation by limiting the diffusivity of radicals [53], which is a likely explanation for the behaviour of the material. Under such conditions, a lower reaction exotherm is predicted and lower temperature than for 1-CuBG-Si. A moderate PS increase after the irradiation period compared to 10-CuBG and 1-CuBG-Si supported this consideration.

Influenced by the low DC and the same elastic modulus as 1-CuBG-Si [5], the PSS values for 5-CuBG-Si were levelled with 14-Si. For both ternary composites, 1-CuBG-Si and 5-CuBG-Si, the polymerisation kinetics data showed a faster transition to the glass phase (higher parameter ‘b’) and an equally high maximum PSS rate, as well as a short time to reach the maximum PSS rate. In contrast, for 5-CuBG-Si, the polymerisation in the glass phase was slower (lower parameter ‘c’), which probably resulted in lower internal stresses and lower final PSS values [46,54]. It is also possible that the proposed Cu-MBGN/silica supraparticles acted as stress-relieving sites due to their weak physical bonds, thus reducing PSS.

In this study, the bioactive control materials 10-BG and 14-BG showed a well-documented reduction in DC [30,32,55,56] despite the high light transmittance [57]. The dose-dependent inhibition of polymerisation is related to the presence of oxides on the surface of 45S5 glass, which interacts with free radicals to form radical oxides that interfere with the polymerisation reaction at room temperature [43]. This behaviour depends on the monomer species, with BisGMA- and Bis-EMA-based composites exhibiting a more pronounced DC decrease and urethane dimethacrylate-based composites achieving clinically acceptable values and depths of cure [57]. This theory was also confirmed in this study, in which the same BisGMA/TEGDMA base was used for all materials. For most of the tested parameters (DC polymerisation kinetics parameters—maximum reaction rate, PS, PSS, and maximum shrinkage rate), materials 10-BG and 14-BG had the lowest values. However, the depth of cure met ISO 4049 requirements.

## 4. Materials and Methods

### 4.1. Materials

We synthesised Cu-MBGN according to a protocol described elsewhere [22]. The fillers used in this study are presented in Table 2. We admixed fillers to form a photoreactive resin mixture composed of bisphenol-A-glycidyldimethacrylate (BisGMA; Merck, Darmstadt, Germany) and triethylene glycol dimethacrylate (TEGDMA, Merck) at a 60/40 ratio, with 0.2 wt.% of camphorquinone (Merck) and 0.8 wt.% of ethyl-4-(dimethylamino) benzoate (Merck). The compositions of the composites are presented in Table 3. The total filler load was kept constant at 65 wt.%.

### 4.2. Light Transmittance

We measured light transmittance using a National Institute of Standards and Technology (Gaithersburg, MA, USA; NIST)-referenced and calibrated MARC^®^ System spectrometer (Bluelight Analytics Inc., Halifax, NS, Canada). We measured radiant exitance with a Bluephase^®^ PowerCure light-curing unit (Ivoclar Vivadent, Schaan, Liechstenstein) using empty compartments in triplicate. The light guide of the curing unit had a 9 mm diameter and emission wavelength maxima at 411 and 450 nm, in the range of 360–540 nm. Radiant exitance measured in the high-power mode amounted to 734 mW/cm^2^. To measure radiant exposure, we placed uncured composite materials in Delrin^®^ moulds (Bluelight Analytics Inc.; d = 6, h = 2 mm) with polyethylene terephthalate (PET) strips covering the top and bottom apertures. We securely positioned the light-curing unit perpendicular to each specimen’s surface using a three-dimensional fixation assembly, and the specimen was light cured for 20 s. We measured real-time irradiance at the bottom of the specimen over 20 s of illumination.

We collected irradiance and radiant exposure individually at a wavelength of 360–540 nm at a rate of 16 records/s. The sensor was triggered at 20 mW. We calculated the radiant exposure by integrating the irradiance with the wavelength at the used exposure time (20 s).

### 4.3. Polymerisation Kinetics

We evaluated polymerisation kinetics using a Fourier-transform infrared (FTIR) spectrometer (Nicolet iS50, Thermo Fisher, Madison, WI, USA) with an attenuated total reflectance (ATR) accessory.

We placed uncured composites (*n* = 5) in custom-made silicone moulds (d = 3, h = 2 mm), covering the ATR diamond and a PET foil on each specimen’s top surface, using the aforementioned light-curing unit as for the light transmittance measurements. Light curing was activated for 20 s using a total energy of 14.7 J/cm^2^. We captured FTIR spectra in real time at a rate of 2 spectra/s for 5 min, with four scans and a resolution of 8 cm^−1^ [54]. We tested five specimens per experimental group (*n* = 5).

We used the changes in the ratios of absorbance intensities of the aliphatic band at 1638 cm^−1^ and the aromatic band at 1608 cm^−1^ to calculate the DC.
(1)$$\mathrm{DC}\;\left(\%\right)=\left[1-\frac{{\left(1638\;{\mathrm{cm}}^{-1}/1608{\;\mathrm{cm}}^{-1}\right)}_{peak\;height\;after\;curing}}{{\left(1638{\;\mathrm{cm}}^{-1}/1608{\;\mathrm{cm}}^{-1}\right)}_{peak\;height\;before\;curing}}\right]\times 100.$$

We plotted the DC data as a function of time and calculated the first derivatives to represent the reaction rate. We plotted the obtained reaction rate as a function of time to determine the maximum reaction rate (R_max_) and the time required to reach the maximum reaction rate (t_max_). Additionally, we evaluated the DC values reached at the end of the 5 min observation period (DC_5min_).

A four-parameter exponential sum function fitted the curves of DC versus time.
y = a × (1 − e^−bx^) + c × (1 − e^−dx^).(2)

We used the four modulation parameters in this equation to describe polymerisation kinetics during the gel phase (parameters a and b) and the glass phase (parameters c and d).

### 4.4. Polymerisation Shrinkage

We measured linear shrinkage in real time using a custom-made linometer, as described previously [43,55,58]. We used eight disc-shaped composite specimens of standardised volume (V = 42 mm^3^, d = 6 mm, h = 1.5 mm) per material. Light curing was performed perpendicularly through the glass plate covering the top of the specimen for 20 s, with a total radiant exposure of 14.7 J/cm^2^. We took measurements in real time for 15 min from the initiation of light curing, and we converted analogue data to digital values using custom-made software.

### 4.5. Polymerisation Shrinkage Stress

We recorded polymerisation shrinkage stress in real time using a custom-made stress analyser and a procedure that was previously described in detail [43,58]. The setup was semi-rigid, with a compliance of 0.4 μm/N that simulated partial shrinkage stress relief, like that of hard dental tissues [48]. Disc-shaped composite specimens (*n* = 8) had the same dimensions as the PS measurements (V = 42 mm^3^, d = 6 mm, h = 1.5 mm). The top of each specimen was bonded to a metal cylinder attached to the load cell (PM 11-K; Mettler, Greifensee, Switzerland), and the base of each specimen was bonded to a glass plate. Both sides had a surface area of 28 mm^2^, amounting to a configuration factor of C = 2.0. The metal cylinder and the glass plate were roughened with 50 μm of aluminium oxide particles, rinsed with deionised water, dried, and silanised using Monobond™ Plus (Ivoclar™ Vivadent). We carried out light curing through the glass plate using the same parameters and radiant exposure as for the PS measurements. The load cell registered the forces originating from PS and collected the data for 15 min at a rate of 5 Hz, with an accuracy of 0.001 N. We divided force values by the specimens’ bonded surface area (28 mm^2^) to obtain shrinkage stress. We plotted the PSS data against time and calculated the first derivatives of these curves to measure the shrinkage stress rate. We calculated the kinetic parameters (maximum shrinkage stress rate and time to achieve maximum shrinkage stress rate) from the plots of the first derivatives against time.

### 4.6. Depth of Cure

We fed uncured composite materials into 4 mm diameter and 10 mm high cylindrical moulds, covered with PET foil, and photopolymerised them with the previously described light-curing unit for 20 s. We then opened the split moulds and removed the unpolymerised excess material. We measured the length of the remaining portion with a calliper and divided the results by two to obtain the depth of cure according to ISO 4049.

### 4.7. Statistical Analysis

We verified the normality of distribution using a Shapiro–Wilk test and normal Q–Q diagrams. We statistically compared the mean values for each variable (light transmittance, degree of conversion after 5 min, maximum reaction rate, time to reach maximum reaction rate, linear shrinkage after 5 min, shrinkage stress after 5 min, maximum shrinkage stress rate, time to reach maximum shrinkage stress rate, and depth of cure) among the materials using one-way ANOVA with Tukey’s post hoc adjustment for multiple comparisons. We performed the statistical analysis using SPSS^®^ (version 25; IBM^®^, Armonk, NY, USA) at an overall α = 0.05.

## 5. Conclusions

In this study, incorporating Cu-MBGN into resin composites increased light scattering but enhanced DC to the highest values, which were measured at a clinically relevant depth of 2 mm. Lower PS and PSS values of the composites, comparable to inert controls and adequate depth of cure indicate their suitability for clinical use. The synergistic effect of silica and 5% Cu-MBGN effectively reduced PSS. Adding bioactive glass 45S5 reduced PS and PSS values, but at the same time also reduced DC. In contrast, adding Cu-MBGN produced PSS values resembling those of inert reference materials, maintaining a high DC.

Considering their previously demonstrated strong mechanical properties and antimicrobial activity, we can conclude that further investigation into multifunctional dental composites is warranted. Material with a combination of 5% Cu-MBGN, silica, and inert Ba-glass micro fillers seems to have the best perspective for future clinical use.

## Figures and Tables

**Figure 1 ijms-23-08195-f001:**
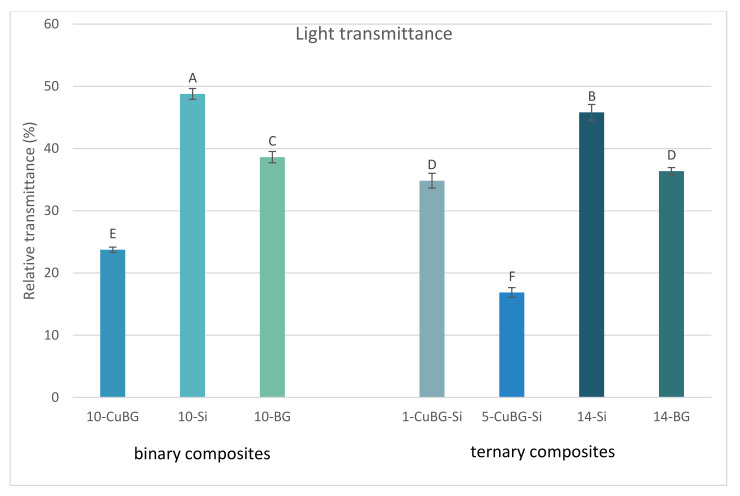
Light transmittance (mean values ± standard deviation) of the tested materials measured at a 2 mm depth. Identical letters denote statistically similar groups.

**Figure 2 ijms-23-08195-f002:**
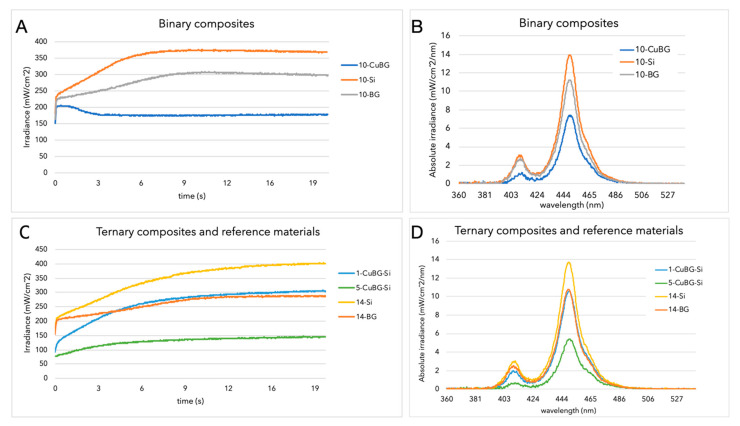
Variations in transmitted irradiance with exposure time and transmitted absolute irradiance for binary (**A**,**B**) and ternary (**C**,**D**) composites. In (**D**), there is a partial overlap of 1-CuBG-Si and 14-BG at the 450 nm peak.

**Figure 3 ijms-23-08195-f003:**
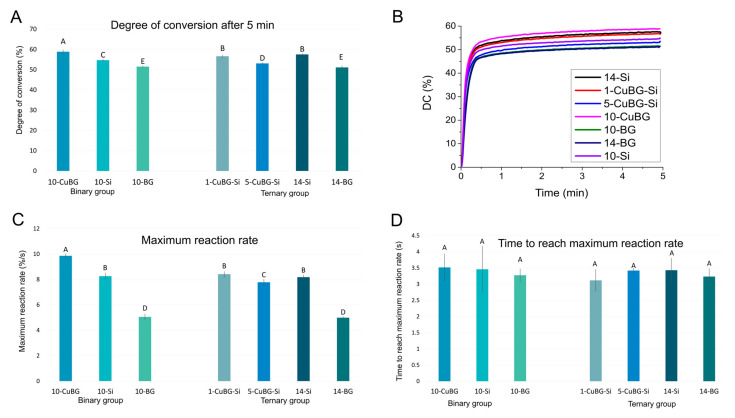
Degree of conversion (**A**), representative curves of the development of the degree of conversion (DC (%)) over 5 min (**B**), maximum reaction rate (**C**), and time to reach the maximum reaction rate (**D**) of tested materials measured at 2 mm depth 5 min after photopolymerisation initiation (mean values ± standard deviation). Identical letters denote statistically similar groups.

**Figure 4 ijms-23-08195-f004:**
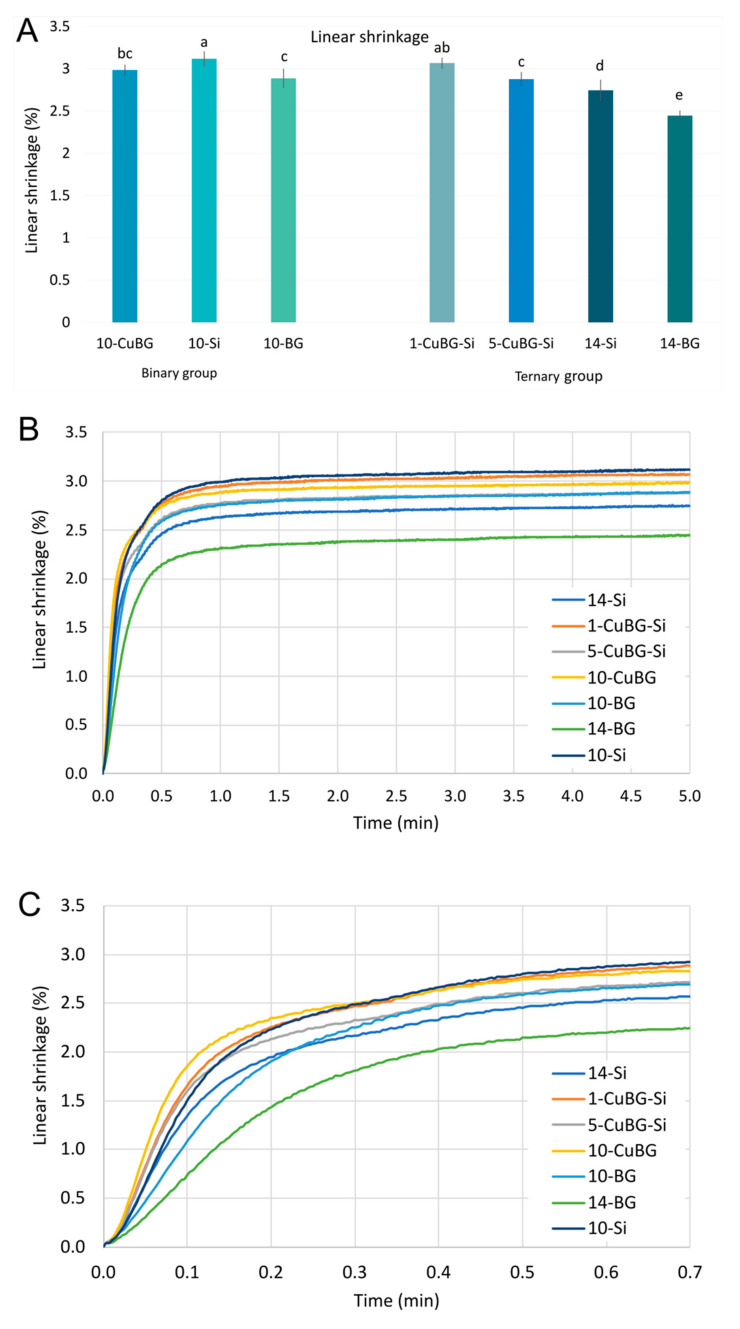
Linear shrinkage: (**A**) final linear shrinkage values after 5 min (mean values ± standard deviation) of binary and ternary composites (identical letters denote statistically similar groups), (**B**) development of linear shrinkage (mean curves) as a function of time over 5 min, and (**C**) development of linear shrinkage (mean curves) as a function of time over 42 s.

**Figure 5 ijms-23-08195-f005:**
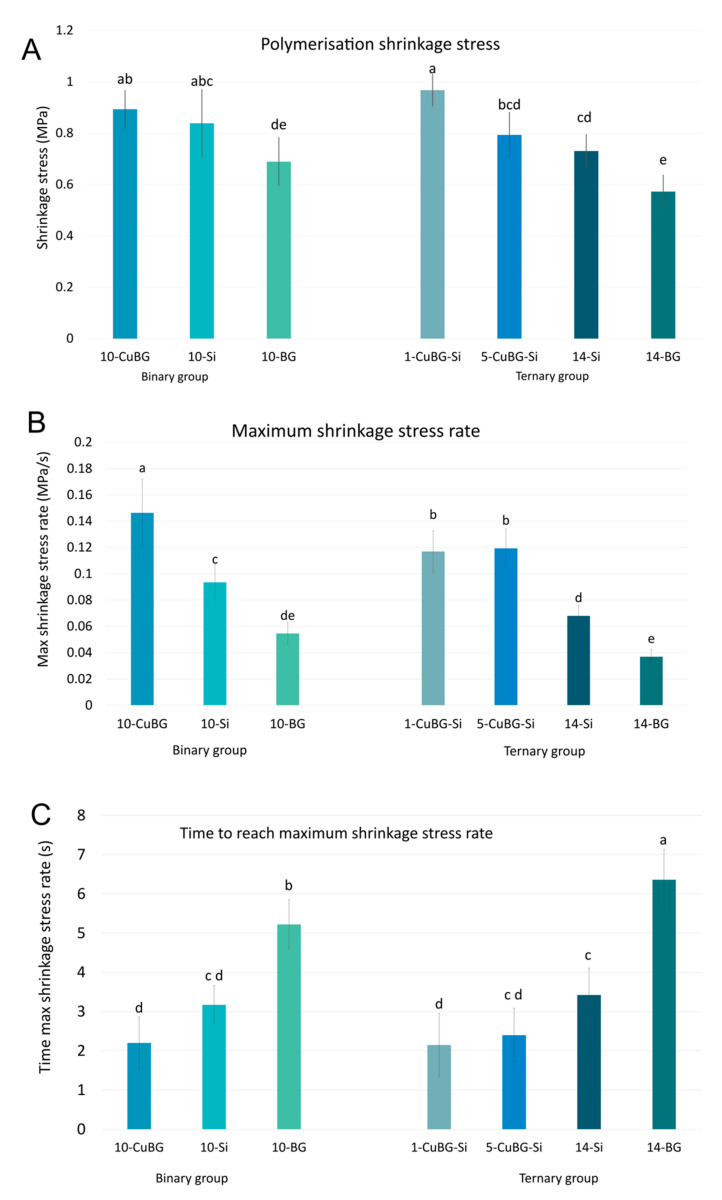
Polymerisation shrinkage stress (mean values ± standard deviation) of binary and ternary composites: (**A**) end values after 5 min measurement, (**B**) maximum polymerisation shrinkage stress rate, and (**C**) time to reach the maximum polymerisation shrinkage stress rate. For all parameters, identical letters denote statistically similar groups.

**Figure 6 ijms-23-08195-f006:**
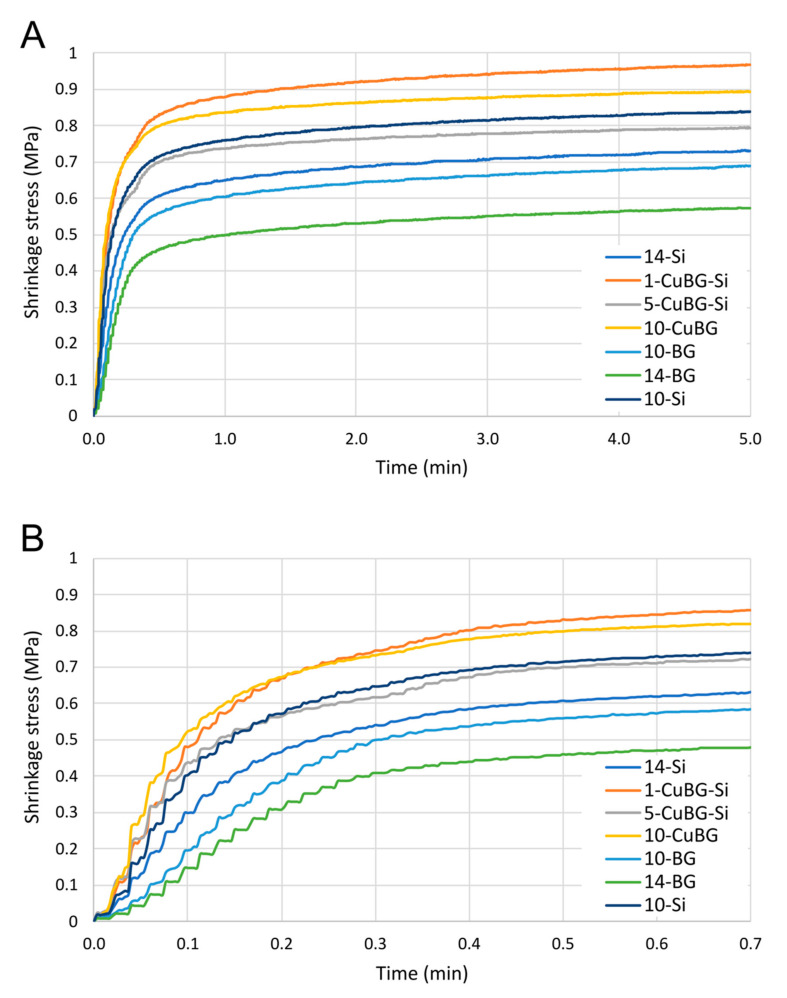
Development of polymerisation shrinkage stress (mean curves) as a function of time over (**A**) 5 min and (**B**) 42 s.

**Figure 7 ijms-23-08195-f007:**
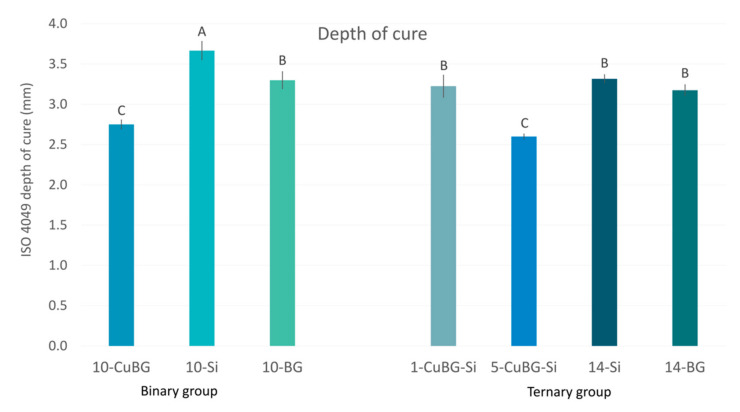
Depth of cure (mean values ± standard deviation) of binary and ternary composites. Identical letters denote statistically similar groups.

**Table 1 ijms-23-08195-t001:** Parameters of polymerisation kinetics (exponential sum function y = a × (1 − e^−bx^) + c × (1 − e^−dx^)).

	a	b	c	d
10-CuBG	63.34	11.63	7.55	0.65
10-Si	59.12	10.56	5.86	0.56
10-BG	55.80	8.00	5.51	0.31
1-CuBG-Si	58.81	11.53	7.57	0.60
5-CuBG-Si	55.92	10.96	7.01	0.66
14-Si	61.44	10.49	7.33	0.51
14-BG	55.75	7.72	5.16	0.33

**Table 2 ijms-23-08195-t002:** Characteristics of fillers used in the present study (data provided by the manufacturers).

Name	Type	Manufacturer/ Product	Composition (wt.%)	Size	Silanisation
Cu-MBGN	Experimental/bioactive	Produced in-house [22]	SiO_2_ 84.8% CaO 9.4% CuO 5.8% *	~100 nm	No
45S5 BG	Commercial/bioactive	Schott, Mainz, Germany G018-144	SiO_2_ 45% Na_2_O 24.5% CaO 24.5% P_2_O_5_ 6%	4.0 μm	No
Ba glass	Commercial/inert	Schott, Mainz, Germany GM27884	SiO_2_ 55.0% BaO 25.0% B_2_O_3_ 10.0% Al_2_O_3_ 10.0%	1.0 μm	Yes 3.2%
Silica	Commercial/inert	Evonik Degussa, Hanau, Germany Aerosil DT	SiO_2_ > 99.8%	12 nm	Yes 4–6%

* Composition determined by inductively coupled plasma atomic emission spectroscopy (ICP-AES), analysis and recalculated to wt.% (data from Zheng et al. [22]).

**Table 3 ijms-23-08195-t003:** Compositions of experimental resin composites (all amounts in wt.%). The total filler load amounted to 65 wt.%.

Group	Material	Resin	Inert Ba Glass Micro Fillers	Silica Nanofillers	Cu-MBGN	45S5 BG
Binary Composites	10-CuBG	35%	55%	-	10%	-
	10-BG			-	-	10%
	10-Si			10%	-	-
Ternary Composites	1-CuBG-Si	35%	51%	13%	1%	-
	5-CuBG-Si			9%	5%	-
	14-BG			-	-	14%
	14-Si			14%	-	-

## Data Availability

The data presented in this study are available on request from the corresponding author.

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
