# Peer review of "Impact of Copper-Doped Mesoporous Bioactive Glass Nanospheres on the Polymerisation Kinetics and Shrinkage Stress of Dental Resin Composites"

_ijms, 2022, doi:10.3390/ijms23158195_

Round 1

Reviewer 1 Report

The manuscript describes  seven composites with Bis-20 GMA/TEGDMA (60/40) matrix and 65 wt% total filler content, added Cu-MBGN or a combination of Cu-MBGN and silanised silica to the silanised barium glass base and examined nine parameters: light transmittance, degree of conversion (DC), maximum polymerisation rate (Rmax), time to reach Rmax, linear shrinkage, shrinkage stress (PSS), maximum PSS rate, time to reach maximum PSS rate, and depth of cure.  The extensive study gives good information on the behaviour of dental resins in the presence of different materials. However, the large number of materials in binary system and ternary system makes the comparison between them complex in spite of the previous hypotheses described.

It is necessary minor revision of manuscript to improve it to achieve the standards of the journal.

The authors should address the following points and revisions:

1. Introduction

- In points (2) and (3) explain more clearly in which terms are no differences between the materials.

2. Materials and methods

- In Figure 2 change the colours to distinguish better the studied materials. Moreover in Figure 2D the material 14-BG is missing.

- In Figure 3 t is not clear whether the degree of conversion is different from %DC. Clarify and improve the explanation.

2. Conclusions

The conclusion is very vague, please specify better which would be the best materials for the dental application.

Reviewer 2 Report

This study continues the previous work by almost the same Authors in ref. [5], which aimed to investigate the effect of Cu-MBGN on composite properties related to polymerisation, and to determine the influence of mesoporous particles on the shrinkage and shrinkage stress of dental composites. As in their previous study [5], the Authors doped resin composites with Cu-MBGN in complex binary or ternary filler systems and standardised Bis-GMA/TEGDMA resin and inert silanised Ba-glass micro fillers (1 μm) for all composites. They found that Cu-MBGN without silica accelerated polymerisation, reduced light transmission, had the highest DC (58.8 ± 0.9 %) and Rmax (9.8 ± 0.2 %/s), but lower shrinkage (3 ± 0.05 26 %) and similar PSS (0.89 ± 0.07 MPa) versus the inert reference (0.83 ± 0.13 MPa). Combined Cu-MBGN and silica slowed the Rmax and achieved a similar DC but resulted in higher shrinkage. However, using a combined 5 wt% Cu-MBGN and silica, the PSS resembled that of the inert reference. The Authors report that,unlike conventional 45S5 bioactive glass, incorporating Cu-MBGN into dental composites provided similar polymerisation properties to the inert reference materials. The paper is very significant and quite original. The methodology is clearly explained, very convincing and sound. The conclusions are well supported by the results. The paper deserves to be published and has a special importance to the Readers. I recommend publishing it as it stands.
